# High Efficacy by GAL-021: A Known Intravenous Peripheral Chemoreceptor Modulator that Suppresses BK_Ca_-Channel Activity and Inhibits *I*_K(M)_ or *I*_h_

**DOI:** 10.3390/biom10020188

**Published:** 2020-01-25

**Authors:** Te-Ling Lu, Zi-Han Gao, Shih-Wei Li, Sheng-Nan Wu

**Affiliations:** 1School of Pharmacy, China Medical University, Taichung City 40402, Taiwan; lutl@mail.cmu.edu.tw; 2Department of Physiology, National Cheng Kung University Medical College, Tainan City 70101, Taiwan; hhelen000111tw@gmail.com (Z.-H.G.); lisway2019vic@gmail.com (S.-W.L.); 3Institute of Basic Medical Sciences, National Cheng Kung University Medical College, Tainan City 70701, Taiwan; 4Department of Medical Research, China Medical University Hospital, China Medical University, Taichung City 40402, Taiwan

**Keywords:** GAL-021, Ca^2+^-activated K^+^ current, large-conductance Ca^2+^-activated K^+^ channel, M-type K^+^ current, pituitary tumor cells, *α-hSlo*

## Abstract

GAL-021 has recently been developed as a novel breathing control modulator. However, modifications of ionic currents produced by this agent remain uncertain, although its efficacy in suppressing the activity of big-conductance Ca^2+^-activated K^+^ (BK_Ca_) channels has been reported. In pituitary tumor (GH_3_) cells, we found that the presence of GAL-021 decreased the amplitude of macroscopic Ca^2+^-activated K^+^ current (*I*_K(Ca)_) in a concentration-dependent manner with an effective IC_50_ of 2.33 μM. GAL-021-mediated reduction of *I*_K(Ca)_ was reversed by subsequent application of verteporfin or ionomycin; however, it was not by that of diazoxide. In inside-out current recordings, the addition of GAL-021 to the bath markedly decreased the open-state probability of BK_Ca_ channels. This agent also resulted in a rightward shift in voltage dependence of the activation curve of BK_Ca_ channels; however, neither the gating charge of the curve nor single-channel conductance of the channel was changed. There was an evident lengthening of the mean closed time of BK_Ca_ channels in the presence of GAL-021, with no change in mean open time. The GAL-021 addition also suppressed M-type K^+^ current with an effective IC_50_ of 3.75 μM; however, its presence did not alter the amplitude of *erg*-mediated K^+^ current, or mildly suppressed delayed-rectifier K^+^ current. GAL-021 at a concentration of 30 μM could also suppress hyperpolarization-activated cationic current. In HEK293T cells expressing *α-hSlo*, the addition of GAL-021 was also able to suppress the BK_Ca_-channel open probabilities, and GAL-021-mediated suppression of BK_Ca_-channel activity was attenuated by further addition of BMS-191011. Collectively, the GAL-021 effects presented herein do not exclusively act on BK_Ca_ channels and these modifications on ionic currents exert significant influence on the functional activities of electrically excitable cells occurring in vivo.

## 1. Introduction

GAL-021 has recently been developed as a novel breathing control modulator thought to preserve respiratory drive and to protect patients from the respiratory impairment resulting from opioids and other modalities; moreover, it notably did not influence analgesia [1,2,3,4,5,6,7]. The studies have also reported that this agent is an experimental drug demonstrated to inhibit Ca^2+^-activated K^+^ channels with big conductance functionally expressed on type 1 cells of the carotid bodies [1,8,9]. However, to what extent this compound perturbs other patterns of voltage-gated ionic currents has not yet been determined.

The big-conductance Ca^2+^-activated K^+^ (BK_Ca_) channels (KCa1.1, KCNMA1, *Slo1*), belonging to family of voltage-activated K^+^ channels, are stimulated by increasing Ca^2+^ concentrations in the cell and by membrane depolarization, and the channel activation of their own accord can then conduct large amounts of potassium ions (K^+^) across the cell membrane. Because of the large conductance, BK_Ca_ channel is also known as maxi- or big-K channel. BK_Ca_ channels, which are functionally expressed in a variety of cells, can play substantial roles in physiological and pathophysiological events including neurotransmitter release, muscle relaxation, and oxygen sensing in chemoreceptor cells [1,8,10,11,12,13,14,15,16,17,18,19]. It is also noted that BMS-204352 (MaxiPost^TM^), an activator of BK_Ca_ channels, could activate a voltage-independent KCNQ4-encoded current [20,21].

The KCNQ gene family encodes five Kv7 K^+^ channel subunits. The KCNQ2, KCNQ3, and KCNQ5 genes encode the core subunits of Kv7.2, Kv7.3, and Kv7.5 channels, respectively, expressed in the nervous system. They are the principal molecular components of the voltage-gated M-type currents (*I*_K(M)_), the magnitude of which can widely regulate neuronal excitability. Once activated, they are characterized by a slow activating and deactivating property [22,23,24,25]. KCNQ2 and KCNQ3 were noted to be mutated in patients with benign familial neonatal convulsions (BFNC). This implies that the activity of M-type (or KCNQx) channels can assist in controlling seizure discharges. Pharmacological targeting of *I*_K(M)_ can be as an adjunctive therapy for various neurological disorders associated with neuronal over-excitability, such as cognitive dysfunction, neuropathic pain, and epilepsy [23,26,27].

Hyperpolarization-activated cationic current (*I*_h_ or funny current [*I*_f_]) is inwardly directed when activated, and therefore increased magnitude of the current depolarizes the membrane potential. *I*_h_ contributes repetitive electrical activity in various types of electrically excitable cells, such as heart cells, neuroendocrine or endocrine cells, or central neurons [28,29,30,31]. This type of current is carried by Na^+^ and K^+^ ions under normal physiological conditions, and the activation of its own accord, then, depolarizes membrane potential, whereby it readily leads the resting potential of the cell to reach the threshold required for action potential generation. The *I*_h_ channels are encoded by the hyperpolarization-activated cyclic nucleotide-gated (HCN) gene family. The HCN gene belongs to the superfamily of voltage-gated K^+^ channels and so far, four HCN channel isoforms (i.e., HCN1-4) exist. Active respiratory neurons were previously reported to functionally express HCN channels [32,33,34]. However, little information is available regarding the underlying mechanism of actions of GAL-021 or other related compounds on different types of ionic currents (e.g., *I*_K(M)_ and *I*_h_) present in endocrine or neuroendocrine cells, or central mammalian neurons, although it was previously reported to suppress the activity of BK_Ca_ channels in type 1 cells of the carotid bodies [1]. Therefore, in this study, we wanted to investigate the ionic mechanisms by which the presence of GAL-021 is able to interact functionally with ion channels, thereby producing any clear modifications on macroscopic ionic currents identified in pituitary GH_3_ cells. Macroscopic or microscopic ionic currents studied include Ca^2+^-activated K^+^ current (*I*_K(Ca)_), BK_Ca_ channels, *I*_K(M)_, *I*_h_, *erg*-mediated K^+^ current (*I*_K(erg)_), and delayed-rectifier K^+^ current (*I*_K(DR)_). Suppressive effects of GAL-021 on *α-hSlo* expressing HEK293T renal epithelial cells were also tested in this study. The findings from these results are useful as they highlight potential mechanisms of ionic action of GAL-021 or related compounds in electrically excitable cells, if similar in vivo observations occur.

## 2. Materials and Methods

### 2.1. Chemicals, Drugs, and Solutions

For this study, GAL-021 (*N*^2^-methoxy-*N*^2^-methyl-*N*^4^,*N*^6^-dipropyl-1,3,5-triazine-2,4,6-triamine or *N*-[4,6-*bis*-*n*-propylamino-(1,3,5)-triazin-2-yl]-*N*,*O*-dimethylhydroxyamine, C_11_H_22_N_6_O) was acquired from MedChemExpress (Everything Biotech, New Taipei City, Taiwan). Flupirtine, ionomycin, 9-phenanthrol, tetrodotoxin, and verteporfin were obtained from Sigma-Aldrich (Merck Ltd., Taipei, Taiwan), BMS-191011, telmisartan, and XE991 were obtained from Tocris (Union Biomed Inc., Taipei City, Taiwan), and dexmedetomidine was from Abbott Laboratories (Taipei City, Taiwan). For cell preparations, all culture media, fetal calf serum, horse serum, l-glutamine, and trypsin/EDTA were acquired from HyClone^TM^ (Thermo Fisher Scientific, Logan, UT), unless stated otherwise, while all other chemicals such as CdCl_2_, EGTA, HEPES, and aspartic acid, were commercially and of reagent grade. In the experiments, we used the twice-distilled water that had been de-ionized through a Millipore-Q system.

The composition of bath solution for GH_3_ cells, that is, normal Tyrode’s solution buffered by HEPES, was as follows (in mM): NaCl 136.5, KCl 5.4, CaCl_2_ 1.8, MgCl_2_ 0.53, glucose 5.5, and HEPES 5.5 titrated to pH 7.4 with NaOH. To measure macroscopic K^+^ currents (i.e., *I*_K(Ca)_, *I*_K(M),_ or *I*_K(DR)_), we filled the patch pipette with a solution (in mM): KCl 140, MgCl_2_ 1, Na_2_ATP 3, Na_2_GTP 0.1, EGTA 0.1, and HEPES 5 titrated to pH 7.2 with KOH. To measure BK_Ca_-channel activity under the inside-out patch configuration, the bath solution contained a high K^+^ solution (in mM): KCl 130, NaCl 10, MgCl_2_ 3, glucose 6, and HEPES-KOH buffer 10 titrated to pH 7.4 with KOH, while the pipette solution contained (in mM): KCl 145, MgCl_2_ 2, and HEPES 5 titrated to 7.4 with KOH. The value of free Ca^2+^ concentration was assessed, assuming that there was a dissociation constant of 0.1 μM for EGTA and Ca^2+^ (at pH 7.2). For example, to provide 0.1 μM Ca^2+^ in the bathing solution, 1 mM EGTA and 0.5 mM CaCl_2_ were added. We commonly filtered the pipette solutions and culture media on the day of use with Acrodisc^®^ syringe filter with 0.2 mm Super^®^ membrane (Bio-Check Lab., Pall Corp., Taipei City, Taiwan).

### 2.2. Cell Preparations

GH_3_ pituitary tumor cells, acquired from the Bioresources Collection and Research Center ([BCRC-60015]; Hsinchu, Taiwan), were maintained in Ham’s F-12 media supplemented with 15% horse serum (*v*/*v*), 2.5% fetal calf serum (*v*/*v*), and 2 mM L-glutamine (Wu et al., 2017). To facilitate cell differentiation, cells were transferred to a serum-free, Ca^2+^-free medium. Under our experimental conditions, cells remained 80% to 90% viable for at least two weeks. The HEK293T cell line was obtained from American Type Culture Collection (CRL-11268; Union Biomed Inc., Taipei City, Taiwan). Cells were grown in Dulbecco’s modified Eagle’s medium supplemented with 2 mM l-glutamine and 10% fetal bovine serum (*v*/*v*). Cells were maintained at 37 °C in a humidified environment of 5% CO_2_ and 95% air incubator.

### 2.3. Transfection

The pCMV6-XL4 vector containing human BK_Ca_-channel pore-forming α-subunit cDNA (α*-hSlo*; NM_002247.1) was obtained from OriGene^TM^ Technologies (Level Biotechnology, Kaohsiung City, Taiwan). The α*-hSlo* gene was recognized to encode a functional BK_Ca_ channel. We transfected the expression plasmid into HEK293T cells for transient expression. Briefly, the expression plasmid was prepared in 150 mM NaCl as a diluent solution. We mixed polyethylenimine (PEI) (ExGen 500; MBI Fermentas, Hanover, MD, USA) and plasmid together and, then, incubated them for 10 min at room temperature for sufficient binding of the plasmid to PEI. We, then, added plasmid-PEI mixture solution to the 24-well plate and centrifuged it at 280× *g* for 5 min. After centrifugation, transfected cells were incubated at 37 °C for an additional 48 h.

### 2.4. Electrophysiological Measurements

Shortly before the measurements, cells (e.g., GH_3_ or HEK293T cells) were harvested and transferred to a homemade recording chamber positioned on the fixed stage of an inverted Olympus fluorescent microscope (CKX-41; Yuan Yu, Taipei City, Taiwan). Cells were put into in normal Tyrode’s solution at room temperature (22 to 25 °C). After cells were left to adhere to the bottom for several minutes, the recordings were performed. Patch-clamp experiments under either whole-cell, cell-attached, or inside-out mode were achieved with either an RK-400 (Biologic, Claix, France) or an Axopatch-200B amplifier (Molecular Devices, Sunnyvale, CA) [16,35]. We fabricated patch pipettes, the resistance of which was around 3 to 5 MΩ, from Kimax-51 borosilicate capillaries (#34500; Kimble; Dogger, New Taipei City, Taiwan) pulled on either a PP-830 vertical puller (Narishige, Major Instruments, New Taipei City, Taiwan) or a P-97 programmable horizontal puller (Sutter Instruments, Novato, CA, USA), and the pipettes were then fire polished with an MF-83 microforge (Narishige). During measurements, the digitized signals, consisting of voltage and current tracings, were stored online at 10 kHz in an ASUSPRO-BU401LG computer (ASUS, New Taipei City, Taiwan) controlled by pCLAMP 10.7 software (Molecular Devices).

### 2.5. Data Analyses

To assess concentration-dependent inhibition of GAL-021 on the amplitude of *I*_K(Ca)_ or *I*_K(M)_, cells were bathed in normal Tyrode’s or high-K^+^, Ca^2+^-free solution, respectively. For studying *I*_K(Ca)_, each cell examined was maintained at 0 mV and 300 ms voltage step from 0 to +50 mV was delivered, whereas for *I*_K(M)_, we held at the level of −50 mV and the depolarizing pulse to −10 mV with 1 sec duration was applied. Current amplitudes measured at +50 mV (*I*_K(Ca)_) or −10 mV (*I*_K(M)_) in response to depolarizing pulses were taken in the control and during the exposure to different concentrations (0.1 to 30 μM) of GAL-021. The concentration required to suppress 50% of *I*_K(Ca)_ or *I*_K(M)_ amplitude was achieved by use a modified form of the Hill equation:(1)$$\mathrm{Relative}\;\mathrm{amplitude}=\frac{{\left[GAL\right]}^{-n_{H}}\times \left(1-a\right)}{{\left[GAL\right]}^{-n_{H}}+IC_{50}^{-n_{H}}}+a$$
where [GAL] represents the different concentrations of GAL-021; n_H_ and IC_50_ are the Hill coefficient and the concentration required for a 50% inhibition of *I*_K(Ca)_ or *I*_K(M)_ amplitude, respectively. Maximal inhibition (i.e., 1-*a*) of *I*_K(Ca)_ or *I*_K(M)_ was also assessed from this equation.

### 2.6. Single Channel Recordings

Unitary BK_Ca_-channel currents were recorded and then analyzed by pCLAMP 10.7 (Molecular Devices). We implemented multi-Gaussian adjustments of the amplitude distributions among channels, in attempts to evaluate the opening events of single-channel currents. Functional independence between channels was verified by comparing the observed stationary probabilities with the values calculated according to the binomial law. The open-state probabilities of the channels were expressed as *N*⋅*P*_O_ which can be evaluated by use of the following form:(2)$$N\cdot P_{O}=\frac{A_{1}+2A_{2}+3A_{3}+\ldots +nA_{n}}{A_{0}+A_{1}+A_{2}+A_{3}+\ldots +A_{n}}$$
where *N* is the number of active channels in the patch examined, *A*_0_ is the area under the curve of an all-points histogram corresponding to the resting (or closed) state, and *A*_1_ to *A*_n_ are the histogram areas reflecting the levels of distinct open state for 1 to *n* channels residing in the patch.

To determine the effects of GAL-021 on the activation curve of BK_Ca_ channels, the upsloping ramp pulses from +20 to +120 mV with a duration of 1 sec were designed and created from pCLAMP 10.7, and, then, they were applied to the membrane patch. The activation curve elicited during ramp pulses were calculated by averaging current traces in response to 20 voltage ramps and dividing each point of mean current by single-channel amplitude of each potential when the leakage component was corrected. The activation curves of the channel taken with or without GAL-021 addition were least-squares fitted with a modified Boltzmann function:(3)$$\mathrm{relative}\;\mathrm{open}\;\mathrm{probability}=\frac{n_{P}}{1+exp\left[\frac{-\left(V-V_{\frac{1}{2}}\right)qF}{RT}\right]}$$
where *n*_P_ is the maximal open probability of BK_Ca_-channel openings in the control at the level of +120 mV; *V*_1/2_ is the voltage at which there is half-maximal activation of the channel; *q* is the apparent gating charge; and *F, R or T* indicates Faraday’s constant, the universal gas constant, or the absolute temperature, respectively.

### 2.7. Statistical Analyses

To assess linear or nonlinear (e.g., sigmoidal or exponential) curve fitting to the dataset was utilized by using OriginPro (OriginLab; Scientific Formosa, Taipei, Taiwan). The experimental results obtained were analyzed and, then, plotted using OriginPro (OriginLab). They were expressed as mean ± standard error of the mean of (SEM). We used the paired or unpaired Student’s *t*-test, or one-way analysis of variance (ANOVA) followed by Bonferroni’s post-hoc test for multiple comparisons, for statistical evaluation of differences among mean values. We used Kruskal–Wallis nonparametric test, if there was possible violation with respect to the assumption of normality underlying ANOVA. *p* Value <0.05 was considered significant, unless stated otherwise.

## 3. Results

### 3.1. Inhibitory Effect of GAL-021 on Ca^2+^-Activated K^+^ Current (I_K(Ca)_) Amplitude Measured from Pituitary GH_3_ Lactotrophs

In an initial stage of electrophysiological recordings, we performed the whole cell configuration of patch-clamp experiments, in order to assess any possible perturbations of GAL-021 on *I*_K(Ca)_ amplitude identified in these cells. The voltage-clamp current recordings were conducted in cells bathed in HEPES-buffered normal Tyrode’s solution into which 1.8 mM CaCl_2_ was put, and the recording pipette was filled with K^+^-containing solution which contained 140 mM K^+^, 0.1 mM EGTA, and 3 mM ATP. As whole-cell model, that is, when membrane patch was broken by suction, was achieved, we maintained the examined cell at the level of 0 mV, a level which was delivered to inactivate other types of outwardly rectifying K^+^ currents [36], and ionic currents were then evoked by a series of voltage pulses ranging between 0 and +80 mV. This type of macroscopic outward currents with a markedly noisy and outwardly rectifying property has been regarded as *I*_K(Ca)_ which is particularly subject to inhibition by tremorgenic mycotoxins (e.g., paxilline or verroculogen) [16,35,37]. As illustrated in Figure 1A,B, the presence of GAL-021 (1 or 3 μM) produced a progressive decrease in *I*_K(Ca)_ amplitude measured at the entire voltage-clamp steps. For example, at the level of +60 mV, the presence of 3 μM GAL-021 was able to decrease *I*_K(Ca)_ amplitude from 418 ± 36 to 124 ± 13 pA (*n* = 8 and *p* < 0.05). After washout of the compound, current amplitude returned to 372 ± 24 pA (*n* = 7 *p* < 0.05). Similarly, as the cells were exposed to GAL-021 (3 μM), the whole-cell (i.e., macroscopic) conductance of *I*_K(Ca)_ measured at the voltages ranging between +40 and +80 mV was significantly decreased to 3.41 ± 0.12 nS from a control value of 5.03 ± 0.18 nS (*n* = 7 and *p* < 0.05).

Concentration-dependent relation of GAL-021-mediated suppression of *I*_K(Ca)_ was further analyzed. As shown in Figure 1C, the relationship between the GAL-021 concentrations and the relative amplitude of *I*_K(Ca)_ was constructed. Notably, this compound could effectively suppress the amplitude of *I*_K(Ca)_ in a concentration-dependent fashion. On the basis of a least-squares fit to the modified Hill equation, the results yielded a concentration required for half-maximal inhibition of 2.33 μM (i.e., IC_50_) and a Hill coefficient of 1.2.

### 3.2. Comparison of the Effect of GAL-021 on I_K(Ca)_ in the Absence and Presence of Diazoxide, Verteporfin, or Ionomycin

We further tested whether the suppressive effect of GAL-021 on *I*_K(Ca)_ was perturbed by diazoxide, a stimulator of ATP-sensitive K^+^ (K_ATP_) channels [38]; verteporfin, an activator of *I*_K(Ca)_ [35]; or ionomycin, a Ca^2+^ ionophore [39]. As shown in Figure 2, when the cells were depolarized from 0 to +50 mV, exposure to GAL-021 (3 μM) significantly suppressed the amplitude of *I*_K(Ca)_. However, further application of diazoxide (10 μM), still in the presence of GAL-021, did not attenuate its amplitude significantly. Conversely, subsequent addition of either verteportin (10 μ M) or ionomycin (10 μM) was found to reverse the *I*_K(Ca)_ amplitude induced by the presence of GAL-021 alone (Figure 2). For example, there was a notable difference in *I*_K(Ca)_ amplitude between GAL-021 (3 μM) alone and GAL-021 (3 μM) plus verteporfin (10 μM) (46 ± 11 pA, in the control *n* = 8, versus 128 ± 19 pA, in the presence of GAL-021 *n* = 8, *p* < 0.05). The current trace suppressed by the presence of GAL seen in GH_3_ cells was, therefore, identified to be an *I*_K(Ca)_, which was sensitive to attenuation by verteporfin or ionomycin, but not by diazoxide.

### 3.3. Modification of GAL-021 on the Activity of BK_Ca_ Channels in GH_3_ Cells

Because *I*_K(Ca)_ is a large and noisy, K^+^ current which is sensitive to intracellular Ca^2+^ and membrane depolarization, and it results predominantly from the opening of BK_Ca_ channels that have been previously described [36,40]. Therefore, to determine whether the inhibitory effect of GAL-021 on *I*_K(Ca)_ is linked to the perturbations on the unitary amplitude, the channel open-state probability, or both, the activity of BK_Ca_ channels present in these cells was measured and analyzed. As shown in Figure 3, under high-K^+^ conditions, the activity of BK_Ca_ channels can be readily observed in an excised inside-out patch. When the detached patch was exposed to GAL-021, the activity of channel openings was progressively decreased (Figure 3). The open-state probability of the channel measured at the level of +60 mV under control condition (i.e., in the absence of GAL-021) was found to be 0.182 ± 0.012 (*n* = 8). We found out that the addition of GAL-021 (10 μM) to the bath medium decreased the channel activity to 0.022 ± 0.002 (*n* = 8 and *p* < 0.01). However, there was no significant difference in the amplitude of the unitary outward current between the absence and presence of GAL-021 (11.4 ± 0.9 pA in the control versus 11.2 ± 0.8 pA in the presence of GAL-021, *n* = 8 and *p* > 0.05). Further addition of cilostazol (10 μM), still in the presence of GAL-021 (10 μM), was effective at reversing GAL-021-mediated suppression of BK_Ca_-channel activity, while that of 9-phenanthrol (10 μM) had minimal effects on it. Cilostazol or 9-phenanthrol have been known to stimulate the activity of BK_Ca_ channels or intermediate-conductance Ca^2+^-activated K^+^ (IK_Ca_) channels, respectively [40,41,42]. Therefore, it is apparent that the presence of GAL-021 can decrease the opening probability of BK_Ca_ channels in these cells, but not that of IK_Ca_ channels.

### 3.4. Effect of GAL-021 on Kinetic Behavior of BK_Ca_ Channels

We further analyzed the effect of GAL-021 on the gating kinetics of these channels because of its inability to modify single-channel amplitude. In excised patches of control cells, open- or closed-time histograms at the level of +60 mV can be fitted with the goodness of fits by a one or two-exponential curve, respectively (Figure 4). The time constant for the open-time histogram was estimated to be 2.12 ± 0.11 msec, whereas those for the fast and slow components of the closed-time histogram were 11.4 ± 0.8 and 56.3 ± 2.1 msec, respectively (*n* = 7). The BK_Ca_ channels identified could exist in at least two resting or closed states. Notably, in detached patches, the addition of GAL-021 (3 μM) to the bath did not result in any modification on mean open time of the channel (2.09 ± 0.11 msec, *n* = 7 and *p* > 0.05), while it did increase the mean closed time to 24.3 ± 1.9 and 98.2 ± 2.5 ms (*n* = 7 and *p* < 0.05). Therefore, these results showed that this compound produced a conceivable increase in channel closed time. Such changes can help to account for its blocking effect on BK_Ca_ channels observed in GH_3_ cells.

### 3.5. Failure of GAL-021 to Alter Single-Channel Conductance of BK_Ca_ Channels

It was further examined whether GAL-021 affects the single-channel conductance of BK_Ca_ channels obtained under voltage-clamp conditions. To construct the plots of current amplitude as a function of membrane potential, we applied the voltage ramp pulses from 0 to +120 mV with a duration of 1 sec at a rate of 0.1 Hz to the inside-out patches. Figure 5A depicts the unitary amplitude versus membrane potential (*I-V)* relationships of BK_Ca_ channels with or without the addition of GAL-021 (3 μM). The single-channel conductance (i.e., unitary amplitude divided by membrane potential) of BK_Ca_ channels achieved from the linear *I-V* relationship in control (i.e., in the absence of GAL-021) was 165 ± 7 pS (*n* = 11) with a reversal potential of 0 ± 1 mV (*n* = 11). The value was not found to differ significantly from that (164 ± 6 pS, *n* = 11 and *p* > 0.05) measured in the presence of GAL-021 (3 μM). As such, GAL-021 produced no significant change in the single-channel conductance of BK_Ca_ channels, but it indeed suppressed the channel activity in these cells, together with a prolongation in mean closed time of the channel.

### 3.6. Effect of GAL-021 on the Activation Curve of BK_Ca_ Channels

Figure 5B illustrates the activation curve of BK_Ca_ channels taken from the absence or presence of GAL-021 (3 μM). In this set of experiments, the activation curves of the channels were obtained with the aid of the voltage ramp protocol. During the single-channel current recordings, the linear ramp pulses were delivered from +20 to +120 mV with a duration of 1 sec. The plots of the relative open probability as a function of membrane potential applied were constructed and then fitted with a modified Boltzmann function as elaborated under the Materials and Methods section. In control, *V*_1/2_ = 66.8 ± 2.3 mV and *q* = 4.3 ± 0.2 *e* (*n* = 7), whereas in the presence of GAL-021 (3 μM), *V*_1/2_ = 79.3 ± 2.6 mV and *q* = 4.1 ± 0.2 *e* (*n* = 7). Consequently, the presence of GAL-021 (3 μM) caused a 45% decrease in the maximal open probability of BK_Ca_ channels and also resulted in a shift of +13 mV in the voltage dependence of the activation curve of the channels. However, there was no measurable effect on the gating charge (i.e., *q*) of the curve occurring during the exposure to GAL-021. The present observations, thus, led us to indicate that GAL-021 could suppress the activity of these BK_Ca_ channels in a voltage-dependent fashion in GH_3_ cells.

### 3.7. Inhibition of M-type K^+^ Current (I_K(M)_) in GH_3_ Cells Produced by GAL-021

In the next set of experiments, we tested if GAL-021 could have any effects on another type of K^+^ current, namely *I*_K(M)_, stated presently in these cells [25,43]. To enhance the degree of *I*_K(M)_, we bathed cells in high-K^+^ solution and filled up the pipette by using K^+^-containing solution. As shown in Figure 6, the addition of GAL-021 (10 μM) resulted in a significant reduction of *I*_K(M)_ amplitude in these cells. Further application of 10 μM flupirtine, but in continued presence of GAL-021, was observed to attenuate GAL-021-induced decrease of *I*_K(M)_. Similarly, the addition of XE991 or dexmedetomidine to the bath effectively suppressed the amplitude of *I*_K(M)_ elicited by 1 sec long membrane depolarization from a holding potential of −50 mV to −10 mV (Figure 6). Flupirtine or XE991 was reported to stimulate or suppress neuronal KCNQ currents, respectively, while dexmedetomidine could have anesthetic-sparing effects and has been previously demonstrated to suppress K^+^ channels [44,45,46,47]. As a result, the data suggested that GAL-021 at a concentration greater than 10 μM could suppress *I*_K(M)_ in GH_3_ cells.

### 3.8. Effect of GAL-021 on Delayed-Rectifier K^+^ Current (I_K(DR)_) in GH_3_ Cells

We further examined whether GAL-021 can perturb another type of K^+^ currents, namely *I*_K(DR)_, enriched in these cells [36]. In these experiments, we bathed cells in Ca^2+^-free, Tyrode’s solution containing 1 μM tetrododoxin and 0.5 mM CdCl_2_. Tetrododoxin or CdCl_2_ was used to block voltage-gated Na^+^ or Ca^2+^ currents, respectively. Within 2 min of exposing cells to 10 μM GAL-021, the *I*_K(DR)_ amplitude was mildly suppressed (Figure 7). For example, as the examined cells were depolarized from −50 to +50 mV, the addition of GAL-021 at a concentration of 10 μM decreased *I*_K(DR)_ amplitude by 22 ± 3% from 847 ± 42 to 663 ± 35 pA (*n* = 7, and *p* < 0.05). Thus, these findings reflect that the presence of GAL-021 mildly suppresses the amplitude of *I*_K(DR)_ in GH_3_ cells. However, previous studies have demonstrated that the *I*_K(DR)_ present in GH_3_ cells could be mediated by several different subtypes of Kv channels, namely Kv1.2, Kv1.4, Kv1.5, Kv2.1, Kv2.2, Kv3.2, Kv4.1, and Kv5.1 [48]. To what extent the GAL-021 action on *I*_K(DR)_ is specific for a unique subset of delayed-rectifier K^+^ channels still remains to be studied further.

### 3.9. Effect of GAL-021 on Erg-Mediated K^+^ Current (I_K(erg)_) in GH_3_ Cells

We further tested the possible perturbations of GAL-021 on another type of voltage-activated K^+^ current, namely *I*_K(erg)_ [49], inherently in GH_3_ cells. To amplify the magnitude of *I*_K(erg)_, we bathed cells in high-K^+^, Ca^2+^-free solution. However, as shown in Figure 8, we were unable to detect any perturbation by this compound on the peak components of deactivating *I*_K(erg)_ measured at the different levels of membrane potential. For example, at the level of −120 mV, no noticeable modification in the peak amplitude of deactivating *I*_K(erg)_ in response to 1 sec membrane hyperpolarization from −10 mV was demonstrated in the presence of 10 μM GAL-021 (498 ± 27 pA in the control versus 49 9 ± 26 pA in the presence of 10 μM GAL-021, *n* = 8 and *p* > 0.05). However, further application of telmisartan (10 μM), still in the presence of 10 μM GAL-021, was able to suppress *I*_K(erg)_ amplitude effectively, as evidenced by a significant reduction of *I*_K(erg)_ to 249 ± 21 pA (*n* = 7 and *p* < 0.05). Telmisartan has previously been demonstrated to suppress *I*_K(erg)_ identified in heart cells [50]. Therefore, as compared to its inhibitory effect on *I*_K(Ca)_, the *I*_K(erg)_ enriched in GH_3_ cells became relatively resistant to be suppression by GAL-021.

### 3.10. Effect of GAL-021 on Hyperpolarization-Activated Cationic Current (I_h_) Recorded from GH_3_ Cells

Another set of experiments was done to test if GAL-021 could have any influence on *I*_h_ identified in these cells [29,30,31,43]. To record *I*_h_ during membrane hyperpolarization, cells were bathed in Ca^2+^-free, Tyrode’s solution and the recording pipette was filled with K^+^-containing solution. As illustrated in Figure 9, the application of 30 μM GAL-021 was observed to be effective at suppressing *I*_h_ amplitude measured at the entire voltage-clamp steps. The activation time course of *I*_h_ in response to long-lasting membrane hyperpolarization concomitantly became slowed in the presence of GAL-021. For example, upon membrane hyperpolarization from −40 to −120 mV with a duration of 2 sec, the GAL-021 (30 μM) addition decreased current amplitude from 263 ± 23 to 77 ± 15 pA (*n* = 8 and *P* < 0.05). After washout of the compound, current amplitude returned to 212 ± 21 pA (*n* = 8 and *P* < 0.05), which implies that the amplitude partially returned to the control level, due possibly to unknown “run-down” process of the current. Likewise, as cells were exposed to GAL-021 (30 μM), the whole-cell conductance of *I*_h_ measured at the voltages ranging between −100 and −120 mV was evidently decreased to 2.81 ± 0.09 nS from a control value of 6.35 ± 0.87 nS (*n* = 8 and *p* < 0.05). In keeping with these data, during the exposure to 30 μM GAL-021, the activation time constant of *I*_h_ during membrane hyperpolarization was raised from 478 ± 25 to 696 ± 37 ms (*n* = 8 and *p* < 0.05).

### 3.11. Inhibitory Effect of GAL-021 on the Activity of BK_Ca_ Channels Expressed in HEK293T Cells Transfected with α-hSlo

In a final set of current recordings, we also wanted to test whether GAL-021 exerts any effects on BK_Ca_-channel activity in HEK293T cells in which *α-hSlo* was transfected. Under our experimental conditions, the *α-hSlo* transfection into HEK293T cells resulted in the appearance of BK_Ca_ channels. Under inside-out recordings, as GAL-021 was applied to the bath, the probability of BK_Ca_-channel openings was drastically reduced (Figure 10). GAL-021 at a concentration of 30 μM evidently decreased open-state probability of the channel from 0.185 ± 0.012 to 0.009 ± 0.002 (*n* = 8 and *p* < 0.01). Subsequent addition of BMS-191011 (10 μM), still in the presence of 30 μM GAL-021, was effective in reversing channel activity suppressed by GAL-021, as evidenced by a significant elevation of channel activity to 0.132 ± 0.011 (*n* = 6 and *p* < 0.05) by further application of BMS-191011. BMS-191011 has been reported to be an activator of BK_Ca_ channels [20]. As such, in keeping with results from the experiments observed in GH_3_ cells, the data strongly suggested that the presence of GAL-021 was capable of suppressing BK_Ca_ channels that would be open in HEK293T cells expressing *α-hSlo*.

## 4. Discussion

The principal findings of this study are as follows: First, in pituitary GH_3_ lactotrophs, the presence of GAL-021 directly inhibits depolarization-elicited *I*_K(Ca)_ amplitude in a concentration-dependent manner; second, the inhibitory effect of GAL-021 on the open-state probability of BK_Ca_ channels occurs in a voltage-dependent manner; third, there is a lengthening in mean closed time of these channels in its presence, albeit no change in mean open time of the channel; fourth, GAL-021 has minimal noticeable effect on *I*_K(erg)_ amplitude, however, it mildly suppresses *I*_K(M)_; fifth, GAL-021 at a concentration greater than 30 μM suppressed the amplitude of *I*_h_; and sixth, GAL-021 suppresses the probabilities of BK_Ca_-channel openings in *α-hSlo*-expressing HEK-293T cells. Collectively, the present observations led us to propose that the effects on various types of membrane ionic currents presented herein could be one of the ionic mechanisms underlying GAL-021-induced actions, if similar in vivo findings occur in different types of electrically excitable cells (e.g., pituitary cells).

GAL-021-mediated suppression of *I*_K(Ca)_ observed in GH_3_ cells was found to be attenuated by further application of either verteporfin or ionomycin, yet not by diazoxide, an activator of K_ATP_ channels [51]. Vertiporfin was recently reported to activate BK_Ca_ channels [35], while ionomycin is a Ca^2+^ ionophore. In keeping with the present observations and previous studies [5,6], the inside-out current recordings revealed the effectiveness of GAL-021 in decreasing the open-state probability of BK_Ca_ channels that would be open. The inhibitory effect of GAL-021 on such channels is conceivably due to the result of its direct binding to the inner surface of the channel. Moreover, the addition of GAL-021 was able to reduce the open-state probability of BK_Ca_ channels in α*-hSlo*-expressing HEK293T cells. The GAL-021-induced decrease of *I*_K(Ca)_ described in this study is, therefore, unlikely to be implicated to its suppressive action on K_ATP_ or IK_Ca_ channels, even though those channels are functionally active in endocrine cells like GH_3_ cells [24,38]. It is also tempting to speculate that GAL-021 has a unique structure to interact functionally with μ-subunit of the BK_Ca_ channel (i.e., KCNMA1 channel) to decrease the amplitude of *I*_K(Ca)_ observed in GH_3_ cells.

In this study, the decrease by GAL-021 in the probabilities of BK_Ca_-channel openings recorded from GH_3_ cells is not attributed to a decrease in single-channel amplitude, because unitary conductance of the channel taken between the absence and presence of GAL-021 apparently failed to differ. A conceivable lengthening in mean closed time (i.e., fast and slow components) of the channels without modification in mean open time could, however, help to account largely for GAL-021-mediated suppression of channel activity. In this scenario, the presence of GAL-021 can prolong the dwelling time of these channels residing in the closed state, along with no profound increase in the duration of individual openings. Moreover, the present observations showed that the presence of GAL-021 resulted in an evident shift of the activation curve of BK_Ca_ channels along the voltage axis to more depolarized potentials, despite no change in the gating charge for such activation, that is, there is no dramatic effect of GAL-021 on the steepness of the channel activation curve. In this regard, the valence of a moving charge was not reduced, nor was profile of the local field, through which other charged residues move, changed. It is also reasonable to propose that this compound can functionally interact with BK_Ca_ channels in a voltage-dependent manner and its inhibitory actions on these channels would be expected to rely on different conditions, such as the pre-existent resting potential, the concentration of GAL-021 applied, or both, assuming that the GAL-021 action in vivo is similar to those in GH_3_ cells shown here.

GAL-021-mediated suppression of BK_Ca_-channel activity could arise from various patterns of channel kinetic behaviors. Its inability to alter single-channel conductance of the channel seen in GH_3_ cells prompts us to indicate that the site(s) of its interaction mainly residing within central pore cavity of the channel are unlikely. However, under inside-out current recordings, in its presence, the fast and slow components in mean closed time of the channel were significantly prolonged, with no clear modification on mean open time. The GAL-021 presence most likely diminishes channel activity by gaining access to its pertinent binding on channel gating. As such, findings from the present results tend to regard GAL-021 as a BK_Ca_ inhibitor characterized by a greater affinity for the channel preferentially in the closed state.

GAL-021 treatment is noted to have a short duration of action after a bolus injection [1,52]. The major effect of GAL-021 on ionic currents (e.g., *I*_K(Ca)_ or *I*_K(M)_) shown here also had a rapid onset of action. The effective IC_50_ required for its suppression of *I*_K(Ca)_ was 2.33 μM, a value that is clinically achievable [6]. Collectively, the above-given results informed us of the notion that the ion channels (i.e., BK_Ca_ [KCNMA1] and M-type [KCNQx] K^+^ channels) would be a relevant therapeutic target for the pharmacological actions of this drug concurring in vivo, although the detailed mechanism of its actions on ion channels remains to be further resolved.

## Figures and Tables

**Figure 1 biomolecules-10-00188-f001:**
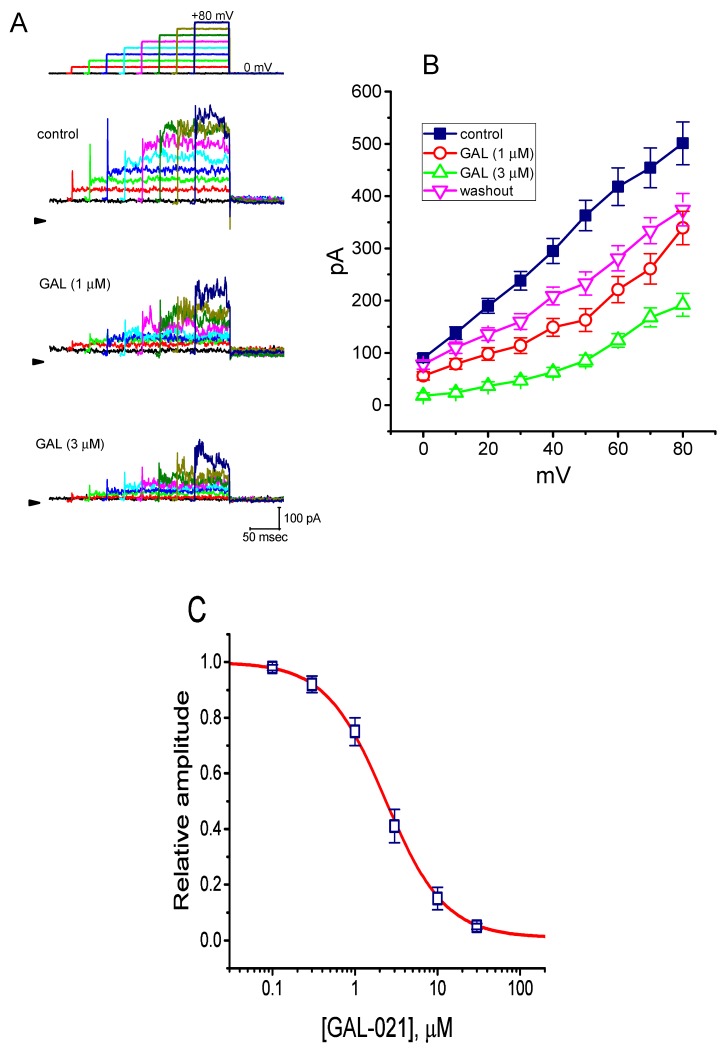
Suppressive effects of GAL-021 on the magnitude of Ca^2+^-activated K^+^ current (*I*_K(Ca)_) recorded from GH_3_ pituitary tumor cells. In this set of whole-cell current recordings, we immersed cells in normal Tyrode’s solution containing 1.8 mM CaCl_2_, and the recording pipette used was filled with a K^+^-containing solution. Current traces are illustrated under voltage-clamp conditions during voltage pulses from 0 to +80 mV in 10-mV increments from a holding potential of 0 mV. (**A**) Superimposed *I*_K(Ca)_ traces elicited during a series of voltage steps (indicated in the uppermost part) obtained under control conditions (upper panel), that is, when GAL-021 was not present, and during cell exposure to 1 μM GAL-021 (middle panel) or 3 μM GAL-021 (lower panel). Arrowhead in each panel depicts the zero-current level, whereas calibration mark in the right lower corner applies to all current traces illustrated. (**B**) Averaged current-voltage (*I-V*) relationships of *I*_K(Ca)_ obtained in the control, during the exposure to 1 or 3 μM GAL-021, and after the washout of GAL-021. Each data point represents the mean ± SEM (*n* = 7 to 9). (**C**) Concentration-response relationship for GAL-021-mediated suppression of *I*_K(Ca)_ amplitude (mean ± SEM and *n* = 8 to 10 for each data point). Each cell was depolarized by 300 ms long voltage step from a holding potential of 0 mV to +50 mV, and current amplitudes at the end of each depolarizing step with or without addition of different GAL-021 concentrations (0.1 to 30 μM) were measured. Continuous line in which data points were overwritten shows least-squares fit of the Hill equation (detailed in the text). The estimated values for IC_50_ and the Hill coefficient were 2.33 μM and 1.2, respectively.

**Figure 2 biomolecules-10-00188-f002:**
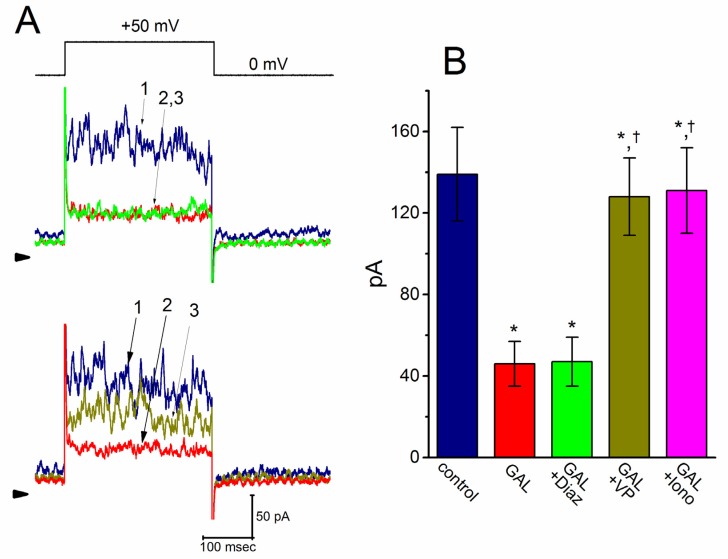
(**A**) Comparisons of effects of GAL-021, GAL-021 plus diazoxide, GAL-021 plus verteporfin, and GAL-021 plus ionomycin on IK(Ca) amplitude recorded from GH3 cells. During these measurements, current amplitude was taken at the end of 300 ms voltage step from a holding potential of 0 mV to +50 mV. Current traces labeled 1 in (**A**) are controls, and those labeled 2 were taken during cell exposure to 3 μM GAL-021, while those labeled 3 were obtained in the presence of 3 μM GAL-021 plus 10 μM diazoxide (upper panel), or in the presence of 3 μM GAL-021 plus 10 μM verteporfin (lower panel). The uppermost part indicates the voltage protocol applied, arrowhead in each panel shows the zero-current level, and calibration mark in the right lower part in (A) applies to all current traces. (**B**) Summary bar graph showing effects of GAL-021, GAL-021 plus diazoxide, GAL-021 plus verteporfin, and GAL-021 plus ionomycin on IK(Ca) amplitude. Current amplitudes were recorded and then measured at the end of each depolarizing pulse from 0 to +50 mV, and each bar represents the mean ± SEM (*n* = 7 to 9). *, Significantly different from control (*p* < 0.05) and †, significantly different from GAL (3 μM) alone group (*p* < 0.05). GAL, 3 μM GAL-021; Diaz, 10 μM diazoxide; VP, 10 μM verteporfin; and Iono, 10 μM ionomycin.

**Figure 3 biomolecules-10-00188-f003:**
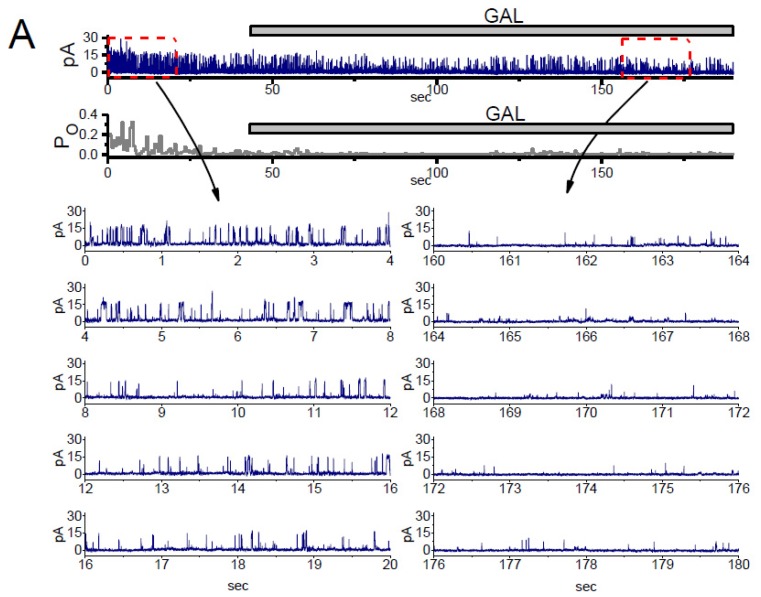
Suppressive effect of GAL-021 on the activity of BK_Ca_ channels identified in GH_3_ cells. In these experiments, cells were immersed in high K^+^ solution containing 0.1 μM Ca^2+^, inside-out current recordings were performed, and channel activity was measured at the level of +60 mV. The upper or lower part in (**A**) shows original channel current traces or the probability of channel openings, respectively. Horizontal bar indicates the addition of 10 μM GAL-021 into the bath. The left (control) or right (during the exposure to GAL-021) current traces in the lowest part of (**A**) corresponds to the expanded records from dashed boxes in the upper part. Channel-opening event is denoted as an upward deflection (i.e., outward current). Of note, there is a progressive decrease in the channel open probability during the exposure to GAL-021 (10 μM). (**B**) Summary bar graph showing effect of GAL-021, GAL-021 plus cilostazol, and GAL-021 plus 9-phenanthrol on the probability of BK_Ca_-channel openings (mean ± SEM and *n* = 6 to 8 for each bar). Inside-out current recordings were taken and the open-state probability of the channel was measured at +60 mV. * Significantly different from control (*p* < 0.05); **, significantly different from control (*p* < 0.01); ^†^ significantly different from GAL-021 (3 μM) alone group (*p* < 0.05); and ^‡^, significantly different from GAL-021 (10 μM) alone group (*p* < 0.05). Cil, 10 μM cilostazol and GAL, GAL-021.

**Figure 4 biomolecules-10-00188-f004:**
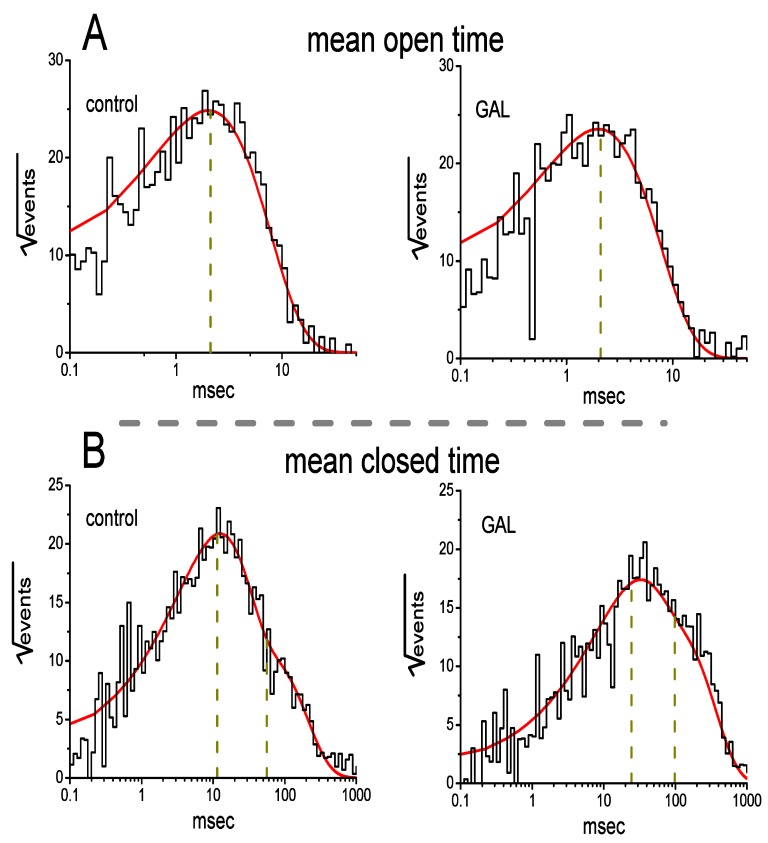
Effect of GAL-021 on mean open (**A**) and closed (**B**) times of BK_Ca_ channels in GH_3_ cells. Inside-out current recordings were made in these experiments and the potential was maintained at +60 mV. Cells were immersed in high-K^+^ solution which contained 0.l μM Ca^2+^. The open- and closed-time histograms under control conditions (i.e., GAL-021 is not present) are illustrated in the left side of each panel, while those acquired during the exposure to 3 μM GAL-021 are in the right side. Of note, the abscissa and ordinate in each panel show the scale of apparent open- or closed-time histograms (msec, in a logarithmic scale) and the square root of the event numbers ($\sqrt{n}$), respectively. Control data were acquired from measurements of 361 channel openings with a total record time of 1 min, whereas those taken in the presence of GAL-021 were taken from 333 channel openings with a total recording time of 2 min. Nonlinear continuous line in each histogram was least-squares fitted by single- or two-exponential function, while the broken line(s) in each lifetime distribution are pointed at the values of the time constants in open or closed state(s). Of note, the fast and slow components in mean closed time of the channel became prolonged in the presence of 3 μM GAL-021, despite its inability to modify mean open time.

**Figure 5 biomolecules-10-00188-f005:**
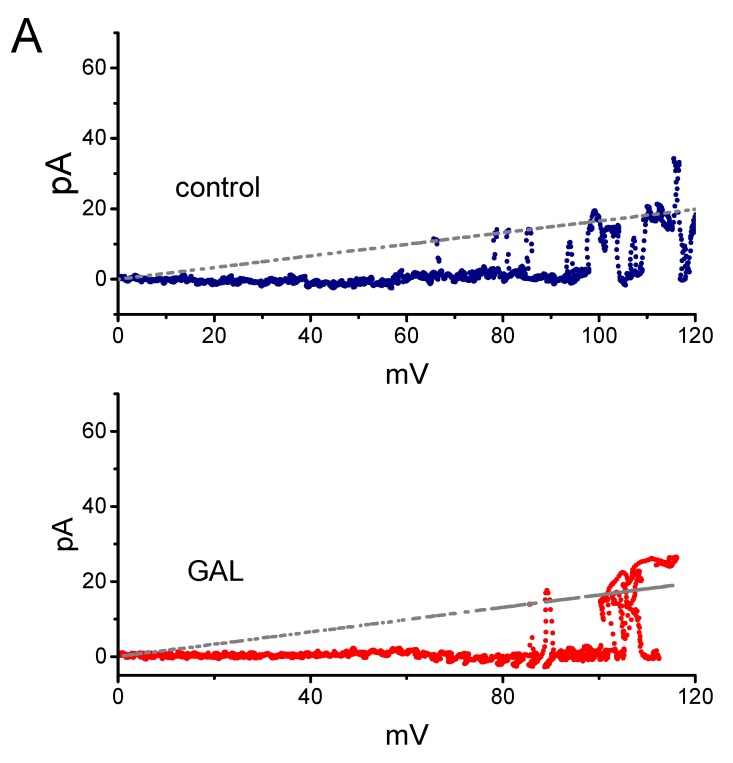
The voltage-dependent effect of GAL-021 on the activity of BK_Ca_ channels identified in GH_3_ cells. The experiments were conducted with symmetrical K^+^ concentration, and we filled the electrode with K^+^-containing solution. Under inside-out current recordings, the potential was maintained at +60 mV and bath medium contained 0.1 μM Ca^2+^. (**A**) Failure of GAL-021 to modify the single-channel conductance of BK_Ca_ channels. The voltage ramp pulses from 0 to +120 mV with a duration of 1 sec were used to measure single-channel conductance taken with or without GAL-021 addition. The straight broken line with a reversal potential of 0 mV represents the *I-V* relations of the channels in the absence (upper) or presence (lower) of GAL-021 (3 μM). (**B**) Effect of GAL-021 on the sigmoid activation kinetics of BK_Ca_ channels. The currents illustrated were activated by applying the upsloping ramp pulses from +20 to +120 mV with a duration of 1 sec. The continuous lines showed Boltzmann fits of the data yielding 66.8 mV for the control (i.e., GAL-021 was not present) and 79.3 mV obtained as the detached patch was exposed to 3 μM GAL-021.

**Figure 6 biomolecules-10-00188-f006:**
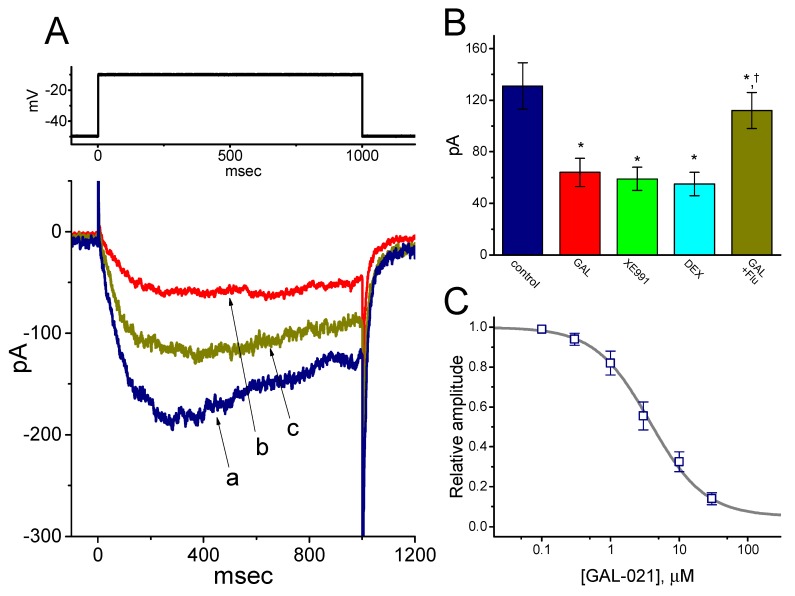
Inhibition by GAL-021 of M-type K^+^ current (*I*_K(M)_) in GH_3_ cells. In this set of experiments, to amplify the current magnitude, cells were immersed in high-K^+^, Ca^2+^-free solution, the examined cell was maintained at −50 mV and, once whole-cell recordings were achieved, the 1 sec long step depolarization to −10 mV was applied. (**A**) Superimposed *I*_K(M)_ traces recorded in the control (a), and during the exposure to 10 μM GAL-021 (b), or to 10 μM GAL-021 plus 10 μM flupirtine (c). (**B**) Summary bar graph showing effects of GAL-021, XE991, dexmedetomidine, and GAL-021 plus flupirtine on *I*_K(M)_ amplitude (mean ± SEM and *n* = 6–9 for each bar). Current amplitudes were measured at the end of each depolarizing command pulse from a holding potential of −50 mV to −10 mV. DEX, 10 μM dexmedetomidine; Flu, 10 μM flupirtine; and GAL, 10 μM GAL-021. ^*^, Significantly different from control (*p* < 0.05) and ^†^significantly different from GAL-021 (10 μM) alone group (*p* < 0.05). (**C**) Concentration-response curve for the GAL-021-induced suppression of *I*_K(M)_ (mean ± SEM and *n* = 6 to 8 for each data point). The smooth sigmoidal curve was fitted with the Hill equation (detailed under Materials and Methods). The IC_50_ value was computed to be 3.75 μM and the Hill coefficient was 1.1.

**Figure 7 biomolecules-10-00188-f007:**
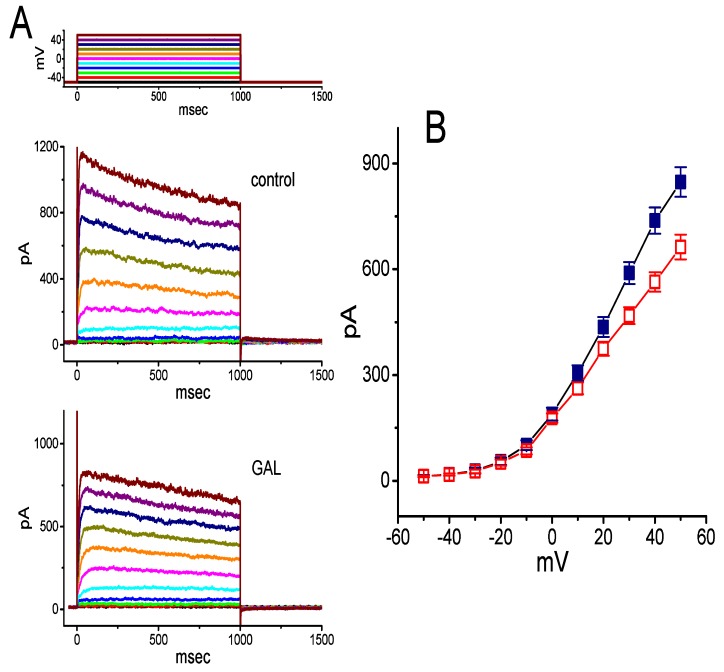
Mild inhibitory effect of GAL-021 on delayed-rectifier K^+^ current (*I*_K(DR)_) in GH_3_ cells. In these experiments, we bathed cells in Ca^2+^-free, Tyrode’s solution which contained 1 μM tetrodotoxin and 0.5 mM CdCl_2_. The holding potential was −50 mV and clamp pulses to a series of voltage steps ranging between –50 and +50 mV with a duration of 1 sec (as indicated in the uppermost part of (A) were applied. (**A**) Representative *I*_K(DR)_ traces recorded in the absence (upper) and presence (lower) of 10 μM GAL-021. (**B**) Averaged *I-V* relationships of *I*_K(DR)_ obtained in the control (■) and during cell exposure to 10 μM GAL-021 (□). Current amplitude was measured at the end of each voltage step, and each data point represents the mean ± SEM (*n* = 7). Of note, GAL-021 at a concentration of 10 μM mildly suppresses *I*_K(DR)_ amplitude.

**Figure 8 biomolecules-10-00188-f008:**
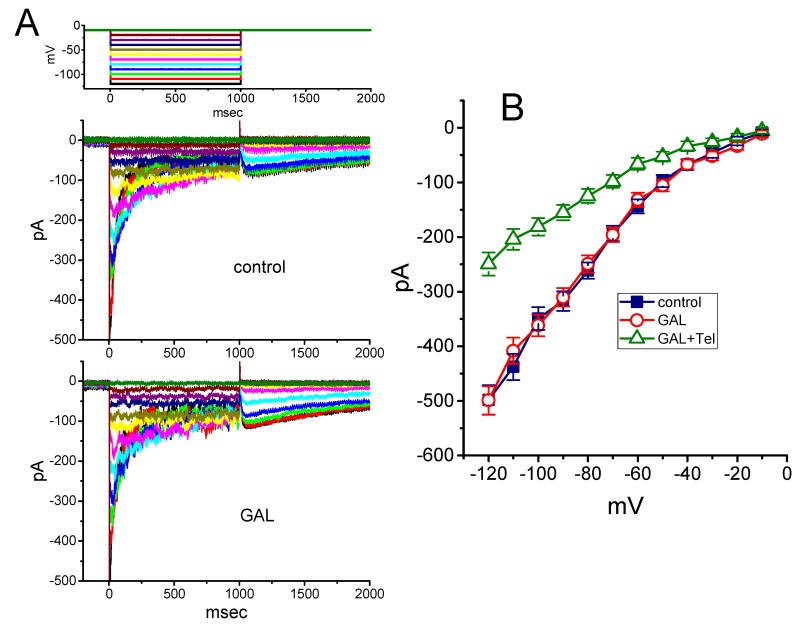
Inability of GAL-021 to suppress *erg*-mediated K^+^ current (*I*_K(erg)_) obtained in GH_3_ cells. We bathed cells in high-K^+^, Ca^2+^-free solution and the pipette used was filled with K^+^-containing solution. (**A**) Superimposed *I*_K(erg)_ traces in the control (upper) and during the exposure to 10 μM GAL-021 (lower). The *I*_K(erg)_ was activated by a series of long-step voltage pulse indicated in the uppermost part of (A). (**B**) Averaged *I-V* relationship of deactivating *I*_K(erg)_ in the absence and presence of GAL-021 (10 μM) or GAL-021 (10 μM) plus telmisartan (Tel, 10 μM). Current amplitude was measured at the beginning of each hyperpolarizing voltage step. Each data point represents the mean ± SEM (*n* = 7 to 8). Note that GAL at a concentration of 10 μM was unable to suppress *I*_K(erg)_ in these cells.

**Figure 9 biomolecules-10-00188-f009:**
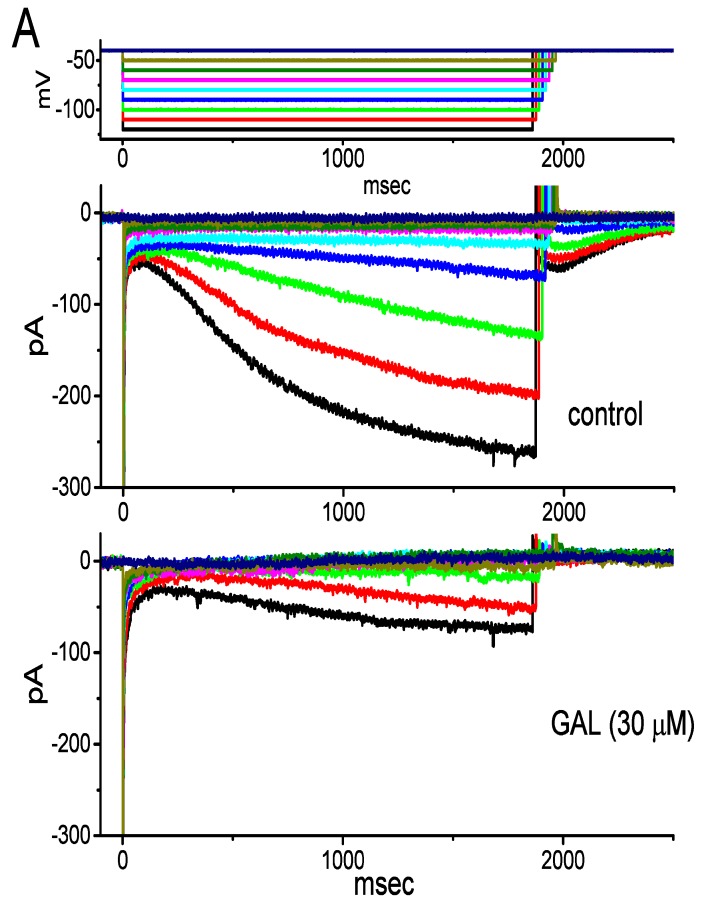
Inhibitory effect of GAL-021 on hyperpolarization-activated cationic current (*I*_h_) identified in GH_3_ cells. In these whole-cell current recordings, cells were bathed in Ca^2+^-free, Tyrode’s solution, and the recording pipette was filled with K^+^-containing solution. (**A**) Representative *I*_h_ traces obtained in the absence (upper) or presence (lower) of 30 μM GAL-021. The uppermost part indicates the voltage-clamp protocol applied. (**B**) Averaged *I-V* relationships of *I*_h_ amplitude obtained in the control (i.e., in the absence of GAL-021), during the exposure to 30 μM GAL-021 and washout of GAL-021 (mean ± SEM and *n* = 8 for each data point). Current amplitude was collected at the end of 2 sec hyperpolarizing potential.

**Figure 10 biomolecules-10-00188-f010:**
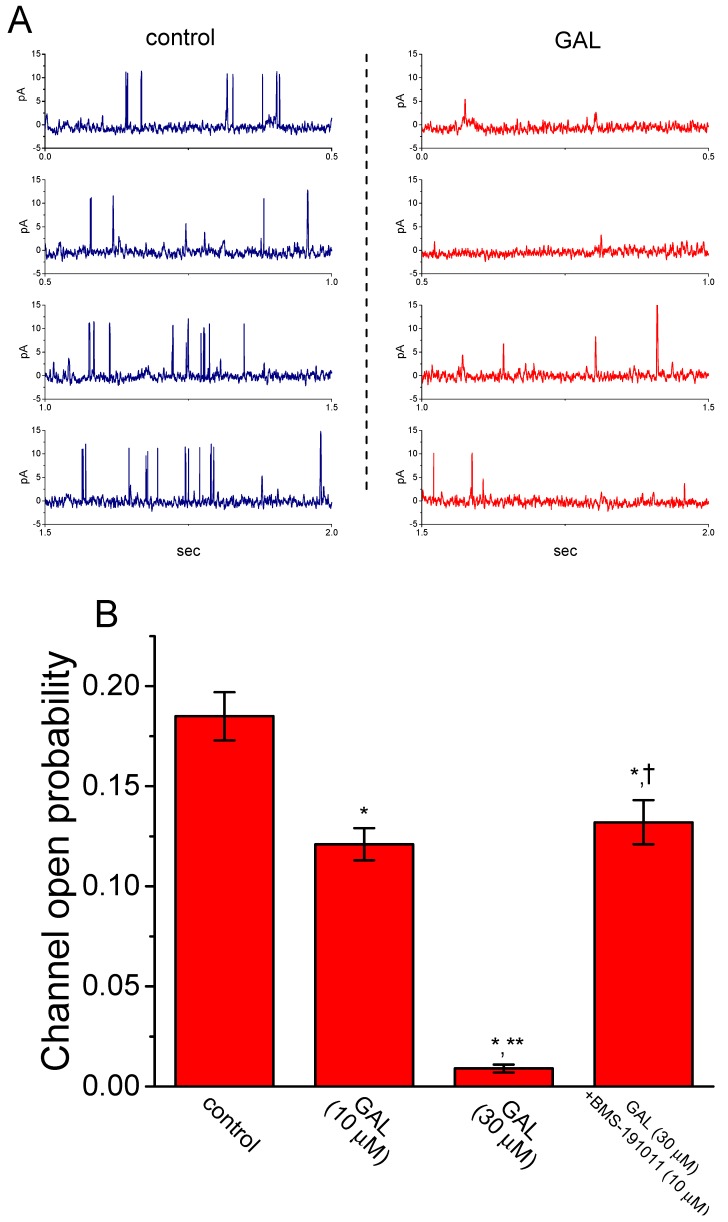
Effect of GAL-021 on BK_Ca_-channel activity in α-*hSlo*-expressing HEK293T cells. Cells were suspended in high-K^+^ solution containing 0.1 μM Ca^2+^, and inside-out current recordings were made at the holding potential of +60 mV. (**A**) Representative BK_Ca_-channel currents obtained in the control (left) and after bath application of 30 μM GAL-021 (right). (**B**) Summary bar graph depicting effect of GAL-021 (10 or 30 μM) and GAL-021 (30 μM) plus BMS-191011 (10 μM) on the open-state probability of BK_Ca_ (or KCNMA1) channels (mean ± SEM and *n* = 6 to 8 for each bar). ^*^, Significantly different from control or GAL-021 (10 μM) alone group (*p* < 0.05); ^**^, significantly different from control (*p* < 0.01); and ^†^, significantly different from GAL-021 (30 μM) alone group (*p* < 0.05).

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
