# Peer review of "High Efficacy by GAL-021: A Known Intravenous Peripheral Chemoreceptor Modulator that Suppresses BKCa-Channel Activity and Inhibits IK(M) or Ih"

_biomolecules, 2020, doi:10.3390/biom10020188_

Round 1

Reviewer 1 Report

The work extended previous finding of GAL-021 (a chemoreceptor modulator) suppression of BK channel activity to show inhibitory mechanism of GAL-021 on BK channels at single-channel levels. The work also includes investigation of GAL-021 on inhibition of M-type K+ and hyperpolarization-activate Ih channel currents. The work helps understand the ionic mechanisms of GAL-021 used as a novel breathing control modulator.

This is the revised manuscript. English grammar was not checked carefully, needs to be double-checked by Word grammar review feature.

Introduction

References are needed for the first and second paragraphs.

Line 89: “conducting” should be “conduct”

Line 145: delete “at resting membrane potential”, using “when activated”. When Ih is activated, membrane potential will change.

Line 152: “A key feature of Ih is its regulation by cyclic nucleotides which” should be replaced by just “Ih channels”.

Line 154-155: HCN channel isoform, not subunits, two different concepts.

Line 1019: “though” should be “through”

Results:

Fig.8: the pulse protocol shown in 8A seems a hyperpolarizing pulse protocol. Isn’t IK(erg) activated by membrane depolarization? Please clarify. Why high K+, Ca2+-free solution was used in the bath? 

IK(erg) is an outward current, why plotted in inward current (negative) in 8B? IK(erg) deactivating current should also be outward current (positive). According to the shape of current traces, it looks more like IK1, the inwardly-rectifying K+ current. Please clarify.

Author Response

        We thank you for the valuable suggestions provided by the reviewer.

        The reply shown below follows the order of questions which were pointed out by the reviewer.

We also upload the manuscript which is the revised version.

Introduction

Thanks for bringing our attention. In the Introduction section, the references in the first and second paragraph were included in the revised manuscript (lines 47-69). “conduct” was corrected (line 60). As indicated by the reviewer, “at resting membrane potential” was replaced with “when activated” (line 83). The sentence was rephrased to “The Ih channels are encoded by the hyperpolarization-activated cyclic nucleotide-gated (HCN) gene family.” (lines 90-91). Thanks for the reviewer’s comment. “subunits” was replaced with “isoforms” (line 92). Goof! We made a mistake. “though” was replaced with “through” (line 639) in the revised manuscript.

Results

In Fig. 8 of the manuscript, the pulse protocol (in Fig. 8A), namely a hyperpolarizing pulse is correct. As cells were bathed in Ca2+-free Tyrode’s solution, IK(erg) could be elicited by long-lasting membrane depolarization. However, in order to amplify the magnitude of IK(erg), we immersed cells in high-K+, Ca2+-free solution. As a result, during a 1-sec membrane hyperpolarization, the deactivating IK(erg) turned out to be inwardly directed, as reported previously (Bauer et al., 1990; Barros et al., 1994; Wu et al., 2000). Hence, the text related to this issue was included in the revised manuscript (line 487-488).

In Figure 8B, the direction of deactivating IK(erg) was indeed inward. The main reason is that cells were bathed in high-K+, Ca2+-free solution for the purpose of amplifying IK(erg), and the value of reversal potential for K+ currents was zero. Hence, as the cell was hyperpolarized from -10 to -100 mV, the IK(erg) virtually became inward. Indeed, the IK(erg) which is sensitive to being blocked by E-4031, an inhibitor of IK(erg), is mildly inwardly rectifying in its biophysical properties of the current, which is distinguishable from those of IK1 (Bauer et al., 1990; Barros et al., 1994; Wu et al., 2000).

References:

Bauer CK, Meyerhof W, Schwarz JR. An inward-rectifying K+ current in clonal rat pituitary cells and its modulation by thyrotrophin-releasing hormone. J Physiol 1990;429:169-189.

Barros F, Villalobos C, Garcia-Sancho J, del Camino D, de la Peña P. The role of the inwardly rectifying K+ current in resting potential and thyrotropin-releasing-hormone-induced changes in cell excitability of GH3 rat anterior pituitary cells. Pflugers Arch 1994;426:221-230.

Wu SN, Jan CR, Li HF, Chiang HT. Characterization of inhibition by risperidone of the inwardly rectifying K+ current in pituitary GH3 cells. Neuropsychopharmacology 2000;23:676-689.

Reviewer 2 Report

The present paper uses electrophysiological methods to adress whether GAL-021, a intraveneous chemoreceptor modulator, affects the Ca2+ activated K+-currents mediated by the BK channel. Using pituitary tumor cells (GH3) and transfected HEK293T cells the author show that GAL-021 supresses BK channel activity. Overall the results presented are interesting. The experiments are performed in a sound way and the results are presented in an easily understandable way. However, the presentation needs some rewriting, in particular the introduction, that lacks a great number of essential references. Every statement in the introduction should be supported by the relevant references. This must be corrected in a revised version of the paper.

Minor changes to the written text

Line 55 remove " after K+ channels

Line 57 conducting should be changed to conduct

Line 62, change the last part to could activate a voltage-independent

Line 89, remove " after the reference

Line 102, change "Materials and Methods" to Materials and Methods (leave out the apostrophe)

Line 106, leave out the apostrophe after the Chemical compound

Line 454, leave out the apostrophe after amplitude of

Author Response

We are grateful to the reviewer’s comments. Notably, the corresponding references in the first and second paragraph of the Introduction section were incorporated into the revised manuscript. The text in the Introduction section was appropriately rephrased. An additional reference was also included in the references section of the revised manuscript (i.e., Wulfsen et al., 2000 [lines 805-807 in the revised manuscript]). The reply shown below follows the order to comments raised by the reviewer.

We also upload the manuscript which is the revised version.

The text in the revised manuscript was corrected and revised. The redundant quotation mark was removed after K+ channels (line 58). Thanks for bringing our attention. “conducting” was corrected to “conduct” (line 60). “… a voltage-independent KCNQ4-encoded current.” was corrected (lines 68). The redundant quotation mark was removed after “….. Weihrauch, 2016)” (line 94). The quotation mark between Materials and Methods was removed (line 108). Thanks! The apostrophe was removed in the revised manuscript (line 111-112). The apostrophe was removed as indicated.

Reviewer 3 Report

The paper is noteworthy and acceptable after a minor revision, which should be focused on following points:

1) The quality of English language should be improved

2) The concentration of free Ca-2+ ions in studies on BK(Ca) channels should be clearly denoted  

3) Did the Authors eliminate the backround noise in recorded BK(Ca) currents during data analysis

4) The inhibitory effect on BK(Ca) currents and Ih currents seem to be partially irreversible. Do the Authors have any idea what is the reason of this partial irreversibility.

5) The shift of V1/2 (Figure 5B) by +13 mV upon an application of GAL-021 should give the V1/2 value of 79 mV (page 20)

6) What types of "delayed rectifier" Kv channels are present in GH3 cells? The Authors should provide pharmacological evidence that currents shown in Figure 7 are indeed due to an activation of "delayed rectifier" channels.

Author Response

Thanks for the comments raised by the reviewer.

        The reply below follows the order of the comments by the reviewer.

We also upload the manuscript which is the revised version.

The quality of English language in the revised manuscript has been appropriately improved. The concentration of free Ca2+ ions in studies on BKCa channels was indicated in the revised manuscript. That is, the text in the revised manuscript (indicated in the red color) appears in lines 130-133, line 332, line 415, and line 561. Yes, in this study, we did minimize the background noise during the recordings of BKCa The representative BKCa-channel tracings are illustrated in Figures 3A and 10A of the revised manuscript. In the experiments on GAL-021 on IK(Ca) amplitude, after washout of the compound, current amplitude returned to 372±24 pA (line 244), indicating that the amplitude almost returned to the control level after the agent being washed out. However, in the experiment on the recordings on Ih, after washout of the GAL-210, current amplitude returned to 212±21 pA, implying that the amplitude partially returned to the control level, due to unknown run-down process of the current (lines 522-523 in the revised manuscript). Good! Thanks for bringing our attention. V1/2 was corrected to 79.3±6 mV (lines 403 and 405 in the revised manuscript). Previous studies have demonstrated that the IK(DR) present in pituitary GH3 cells could be mediated by several different subtypes of KV channels, namely KV2, KV1.4, KV1.5, KV2.1, KV2.2, KV3.2, KV4.1 and KV5.1 (Wulfsen et al., J Neuroendocrinol 2000;12:263-272). The text related to this issue was included in the revised manuscript (lines 467-471). Hence, an additional reference was quoted in the revised manuscript (lines 805-807).